# Antibiotic Susceptibility Profiles of Bacterial Isolates Recovered from Abscesses in Cattle and Sheep at a Slaughterhouse in Algeria

**DOI:** 10.3390/microorganisms12030524

**Published:** 2024-03-05

**Authors:** Chahrazed Yousfi, Saoussen Oueslati, Dina Daaboul, Delphine Girlich, Alexis Proust, Chafia Bentchouala, Thierry Naas

**Affiliations:** 1Centre Hospitalo-Universitaire Ben Badis, Service de Microbiologie, Constantine 25000, Algeria; chahrazedyousfi06@gmail.com (C.Y.); c.bentchouala@yahoo.fr (C.B.); 2Institut des Sciences Vétérinaires, Université des Frères Mentouri Constantine 1, Constantine 25000, Algeria; 3Team ReSIST, UMR1184, INSERM, School of Medicine, OI Healthi, Université Paris-Saclay, CEA, 94270 Le Kremlin-Bicêtre, France; oueslati.saoussen@gmail.com (S.O.); dinadaaboul58@gmail.com (D.D.); delphine.girlich@universite-paris-saclay.fr (D.G.); 4Bacteriology-Hygiene Unit, Bicêtre Hospital, APHP Paris-Saclay, 94270 Le Kremlin-Bicêtre, France; 5Department of Hormonal Biochemistry, Hôpital de Bicêtre, Assistance Publique-Hôpitaux de Paris, 75610 Le Kremlin-Bicêtre, France; alexis.proust@aphp.fr; 6Faculté de Médecine, Université Salah Boubnider Constantine 3, Constantine 25000, Algeria; 7French National Reference Center for Antibiotic Resistance: Carbapenemase-Producing Enterobacterales, 94270 Le Kremlin-Bicêtre, France

**Keywords:** abscesses, bacteria, antimicrobial resistance, cattle, sheep, slaughterhouse

## Abstract

Abscesses represent the most prominent emerging problem in the red meat industry, leading to great economic constraints and public health hazards. Data on etiological agents present in these purulent lesions in Algeria are very scarce. The aim of this study was to identify the bacteria responsible for these abscesses and to determine their antibiotic susceptibility profiles. A total of 123 samples of abscesses from 100 slaughtered sheep and 23 slaughtered cattle were cultured in several media. A total of 114 bacterial isolates were cultured from 103 abscesses. Bacteria were identified using MALDI–TOF, and antibiotic susceptibility was determined by the disk diffusion method on Mueller–Hinton agar. A total of 73.6% (n = 84) corresponded to Enterobacterales, of which four were multidrug-resistant (MDR). These isolates, together with *Staphylococcus aureus*, coagulase negative *Staphylococci*, and seven randomly chosen susceptible *Escherichia coli* isolates, were further characterized using WGS. Resistome analysis of the four MDR Enterobacterales isolates revealed the presence of OXA-48 carbapenemase in two *Klebsiella pneumoniae* ST985 and one *E. coli* ST10 isolates and a CTX-M-15 ESBL in one *E. coli* isolate ST1706. Two coagulase-negative *Staphylococci* isolates were found to carry the *mecA* gene. WGS showed the presence of different resistance genes and virulence genes. Our study revealed 5% of MDR Enterobacterales (including ESBLs and carbapenemases) identified from abscesses, thus urging the need for abscess monitoring in slaughterhouses.

## 1. Introduction

An abscess can be defined as an accumulation of pus surrounded by fibrous tissue. They can occur anywhere in the body where pyogenic bacteria can establish and multiply. Among the common causes of abscesses in cattle and sheep are the following:

Injuries to the feet, such as puncture wounds, bruising, and abscesses, which can result from excessive wear of moist feet on rough and abrasive flooring and poor needle practice, leading to abscesses at injection sites. Caseous lymphadenitis (CL) in sheep and goats is caused by *Corynebacterium pseudotuberculosis* and results in abscesses of peripheral and internal lymph nodes. In cattle, skin abscesses may occur at vaccination sites when vaccination is performed under suboptimal conditions, while liver abscesses are often associated with acidosis in fattening animals [1].

Abscesses are responsible for tremendous economic losses at the farm with a decline in the market value of animals, through a decrease in animal reproductive and productive efficiency, and variable mortality rates. Thus, during the inspection process of production in slaughterhouses, if the carcass presents a problem that could compromise food safety, a total or partial condemnation may occur. This condemnation represents financial losses for the production lines [2,3,4]. In addition, subcutaneous abscesses may be ruptured during the skinning process, and the bacteria responsible for the infection (zoonotic bacteria) may contaminate the surface of the carcass [5,6,7], which represents a public health hazard to slaughterhouse workers and meat inspectors as well as consumers [8].

Many species of zoonotic bacteria may be involved in the etiology of abscesses [6], including *Staphylococcus* spp., *Streptococcus* spp., *Corynebacterium* spp., *Pasteurella* spp., *Pseudomonas aeruginosa*, *Escherichia coli*, and other Gram-negative rods [9,10]. In many instances, samples showed more than one species of bacteria isolated from a single abscess [11].

Local antibiotics (including infusion into the abscess) tend to be more effective than systemic antibiotics. Many antibiotics cannot easily penetrate the capsule of an abscess and/or may not be effective in the abscess environment due to changes in pH and other factors in the pus [12]. In addition, antibiotic choices are limited in food animals, and strict adherence to withdrawal times is required to protect food safety. Most wound infections contain multiple bacterial species and should be cultured to determine optimal antibiotic therapy if needed. Therefore, identification of the specific agents involved in the abscess is necessary to adequately implement effective prophylactic or treatment strategies that could lead to the reduction of microbiological pressure at the farm level and eliminate risk factors [13,14]. 

In Algeria, a few studies were carried out on the epidemiological aspect of the abscesses, but none of these studies addressed the bacteriological aspects and the implication of micro-organisms in relation to their zoonotic potential as well as their antibiotic-resistance profiles and their molecular mechanisms of multidrug resistance [13,15]. The present work was designed to analyze the characteristics of bacteria growing from abscesses in slaughtered cattle and sheep at abattoirs in the Constantine region, northeast of Algeria.

## 2. Materials and Methods

### 2.1. Origin of Isolates

The study was conducted during the period from February 2019 to March 2020 in the largest slaughterhouse in the Constantine region, northeast of Algeria. During the study period, 677 cattle and 978 sheep (N = 1 655) were slaughtered and routinely inspected. The population of both cattle and sheep was predominantly male (561 vs. 116 in cattle and 975 vs. 3 in sheep, for males and females, respectively). A total of 123 abscess lesions were recorded (n = 23, 1.39%) and (n = 100, 6.04%) in cattle and sheep, respectively.

An information sheet was systematically established for each slaughtered animal, including sex, age, and weight status. The localization of examined abscesses in sheep were crural region, liver, lung, lymph node, udder, neckline, peritoneum, precrural region, prescapular region, sternum, cutaneous, and testicles, while in cattle: lung, liver, peritoneum, lymph node, and pericardium.

Intact abscesses from slaughtered cattle and sheep were excised individually, placed in sterile plastic bags, labeled, placed in ice-filled coolers, and transported to the microbiology laboratory of the University hospital Ibn Badis of Constantine, Algeria, for further characterization. 

### 2.2. Microbiological Methods

The abscesses were opened by grasping the surface of the abscesses with a hot spatula and incising the capsule with a sterile scalpel. Observations were made of the abscess size and the consistency, color, and odor of the exudate. A loopful of the material contained in the abscess was streaked directly onto different culture medias including Trypticase Soy Agar, Columbia agar supplemented with 5% of sheep blood, Mac Conkey agar, and mannitol salt agar (BioMérieux, Marcy L’Etoile, France). The incubation was made in aerobic conditions at 37 °C for 24 h. Growing bacterial colonies were subcultured separately on the appropriate media to obtain pure cultures (after another incubation at 37 °C for 24 h).

Bacterial identification was carried out through biochemical galleries (API, bioMérieux), at the Microbiology laboratory of the University hospital Ibn Badis, and subsequently confirmed using MALDI–TOF: matrix-assisted laser desorption ionization–time of flight (Biotyper, Bruker, Hannover, Germany) at the University hospital Bicêtre, Le Kremlin-Bicêtre, France.

### 2.3. Antimicrobial Susceptibility Testing and MIC Determination

All isolates were submitted to susceptibility testing against antimicrobial agents using the Kirby–Bauer disc diffusion assay on Muller–Hinton agar, and the results were interpreted according to the European Committee on Antimicrobial Susceptibility Testing guidelines CA-SFM (Comité de l’antibiogramme de la Société Française de Microbiologie) as updated in 2022 (http://www.eucast.org (accessed on 24 January 2024)). 

The antibiotic disks (bioMérieux) for bacteria belonging to Enterobacterales order were as follows: amoxicillin-clavulanic acid (AMC, 20/10 μg), ampicillin (AMP, 10 μg), amoxicillin (AMX, 25 µg), piperacillin (PIP, 100 µg), ticarcillin (TIC, 75 µg), Temocillin (TEM, 30 µg), cefazolin (CZ, 30 µg), cefoxitin (FOX, 30 µg), cefotaxime (CTX, 30 µg), ceftazidime (CAZ, 30 µg), aztreonam (ATM, 30 µg), imipenem (IPM, 10 µg), ertapenem (ETP, 30 µg), ciprofloxacin (CIP, 5 μg), nalidixic acid (NA, 30 μg), gentamicin (GM, 10 μg), kanamycin (K, 30 µg), amikacin (AN, 30 μg), tetracycline (TET, 30 μg), chloramphenicol (C, 30 µg), fosfomycin (FOS200, 200 µg), and trimethoprim/sulfamethoxazole (SXT, 1.25 + 23.7 µg).

*Aeromonas* spp. isolates were tested towards ceftazidime (CAZ, 30 µg), cefepime (FEP, 30 µg), aztreonam (ATM, 30 µg), ciprofloxacin (CIP, 5 μg), levofloxacin (LEV, 5 μg), and trimethoprim/sulfamethoxazole (SXT, 1.25 + 23.7 µg).

Gram-positive *cocci* were tested against penicillin (P, 10 Units), oxacillin (OX, 1 μg), cefoxitin (FOX, 30 µg), gentamicin (GM, 10 μg), kanamycin (K, 30), Tobramycin (TOB, 10 µg), erythromycin (ERY, 15 μg), clindamycin (CM, 2 μg), chloramphenicol (C, 30 μg), tetracycline (TE, 30 μg), Tigecycline (TGC 15 µg), ciprofloxacin (CIP, 5 μg), ofloxacin (OFX, 5 μg), Levofloxacin (LEV, 5 µg), Quinupristin/dalfopristin (QD, 15 μg), Linezolid (LNZ, 30 µg), trimethoprim/sulfamethoxazole (SXT, 1.25 + 23.7 µg), Rifampin (RA, 5 µg), fusidic Acid (FA, 10 μg), and Nitrofurantoin (F, 300 µg). 

Finally, rod-shaped Gram-positive isolates were tested against meropenem (MER, 10 µg), imipenem (IPM, 10 µg), ciprofloxacin (CIP, 5 μg), erythromycin (ERY, 15 μg), clindamycin (CM, 2 μg), vancomycin (VAN, 30 µg), and Linezolid (LNZ, 30 µg). 

Gram-negative isolates resistant to expanded spectrum cephalosporins and/or to carbapenems were further characterized by broth microdilution method using customized Sensititre plates (Thermo Fisher Scientific, Les Ulis, France). The minimum inhibitory concentrations (MICs) of beta-Lactams, ciprofloxacin, levofloxacin, and tobramycin were determined and interpreted using the EUCAST guidelines. The detection of a carbapenem-hydrolysis was carried out using the homemade Carba NP, and the presence of one of the 5 main carbapenemases was confirmed by the NG-Test CARBA 5 Lateral Flow ImmunoAssay (NG Biotech, Guipry, France) as previously described [16]. 

### 2.4. Whole-Genome Sequencing and Bioinformatic Analysis

Total DNA of twenty bacterial isolates were extracted using the PureLink™ Genomic DNA Mini-Kit (ThermoFisher Scientific, Les Ulis, France) following the manufacturer’s instructions and stored at −20 °C. DNA libraries were prepared using the NEBNext Ultra II FS DNA Library Prep Kit for Illumina (New England Biolabs, Evry, France) according to the manufacturer’s instructions, and run on a NextSeq 500 sequencer (Illumina, Évry-Courcouronnes, France) to generate paired-end 150-bp reads, as previously described [16]. Raw WGS data were assembled de novo using the CLC genomics 10.2 program (Qiagen, Les Ulis, France), and the genomes were analyzed online using software available at the center for genomic epidemiology-CGE (https://cge.food.dtu.dk/ (accessed on 24 January 2024)). The latter included MLST 2.0 software to determine the sequence types (ST), ResFinder 4.1 to determine the acquired resistome, PlasmidFinder 2.1, to identify known plasmid replicon types and VirulenceFinder 2.0. for the presence of potential virulence genes [16]. Additionally, the virulence factor database (VFDB) (http://www.mgc.ac.cn/VFs/main.htm (accessed on 20 December 2023)) was also used to search for virulence factors. Reference plasmid sequences were retrieved from the NCBI database (https://www.ncbi.nlm.nih.gov (accessed on 24 January 2024)). Contigs carrying carbapenemase genes were mapped to reference plasmids, using CLC genomics 10.2 program (Qiagen). Mutations in the quinolone-resistance-determining region (QRDR) of *gyrA* and *parC* were also analyzed.

### 2.5. Statistical Analysis

Collected data were statistically treated using Microsoft Excel 2019 and STATA (version 11.1). Pearson’s chi-square test and Fisher’s exact test, as appropriate, were applied to analyze the categorical variables. A two-tailed *p*-value < 0.05 was considered statistically significant.

### 2.6. Nucleotide Sequence Accession Number

The whole-genome sequences generated in the study have been submitted to the Genbank nucleotide sequence database under Bioproject PRJNA948715.

## 3. Results

### 3.1. Abscess Characteristics

During the study period, 677 cattle and 978 sheep (N = 1655) were slaughtered and routinely inspected. The population of both cattle and sheep was predominantly male (561 vs. 116 in cattle and 975 vs. 3 in sheep, for males and females, respectively). A total of 123 abscess lesions were recorded (n = 23, 1.39%) and (n = 100, 6.04%) in cattle and sheep, respectively. A total of 123 abscess lesions were recorded from the 1655 slaughtered animals, giving an overall prevalence of 7.43%. This prevalence was significantly lower in cattle (n = 23, 1.39%) than in sheep (n = 100, 6.04%) (*p* < 0.001). 

In sheep, 97 (97%) and 3 (3%) of the abscesses were recorded from males and females, respectively, while, in cattle, abscesses recorded from males and females were 19 (82.61%) and 4 (17.39%), respectively. The abscesses were mostly recorded from young animals ((N = 115, 93.5%); sheep (n = 97, 78.87%), and cattle (n = 18, 14.63%)) (Table 1).

Abscesses in sheep were more frequently located in lymph nodes (n = 27, 31.8%), prescapular region (n = 17, 19.8%), and lung (n = 28, 21.2%) (Figure 1A). In cattle, abscesses were frequently located in the lungs (n = 8, 44.4%) and liver (n = 6, 33.3%) (Figure 1B). 

Finally, the characteristics of abscesses vary according to the bacterial species involved, as illustrated in Table 2.

### 3.2. Bacterial Isolates

Of the 123 abscesses analyzed, 103 gave a positive culture (18 in cattle and 85 in sheep), of which 114 bacterial isolates were identified (91 from sheep and 23 from cattle) and further characterized. Bacterial identification revealed Enterobacterales (*Escherichia coli*, *Klebsiella pneumoniae*, *Klebsiella oxytoca*, *Morganella morganii*, *Citrobacter brackii*, *Citrobacter freundii*, *Enterobacter* spp., *Serratia marcescens*, *Proteus vulgaris*, *Lelliottia* spp.) (n = 84, 73.6%), *Staphylococcus* spp. (n = 15, 13.1%), *Aeromonas* spp. (n = 13, 11.4%), and *Bacillus* spp. (n = 2, 1.9%) (Table 3). *Escherichia coli* represented 61.3% of the isolated bacteria, and 83% of the Enterobacterales. 

In the 11 abscesses in which 2 species had been recovered (6 in sheep and 5 in cattle), the different combinations were as follows: *E. coli*/*Aeromonas* spp. (n = 5), *E. coli/S. aureus* (n = 5), and *E. coli*/*Morganella morganii* (n = 1). 

### 3.3. Antibiotic Susceptibility Testing

Different rates of resistance were observed for the Enterobacterales isolates, with resistance to amoxicillin, ticarcillin, ampicillin, amoxicillin-clavulanic acid, and tetracycline being the highest rates recorded (Figure 2). 

Three isolates with reduced susceptibility to carbapenems were identified. Two *K. pneumoniae* isolates (O103B2 and O103B1) had MICs of 2 µg/mL, 1 µg/mL, and 0.5 µg/mL for ertapenem, imipenem, and meropenem, respectively, and one *E. coli* isolate (O103A10) displayed MICs of 0.5 µg/mL for both ertapenem and imipenem, and 0.12 µg/mL for meropenem (Table 3). These isolates presented additionally high MICs for temocillin 512 µg/mL and 256 µg/mL, for *E. coli* O103A10 and *K. pneumoniae* O103B2 and O103B1, respectively (Table 4). These isolates gave positive results using the Carba NP, suggesting the likely presence of a carbapenem-hydrolyzing enzyme. The NG-Test CARBA 5 confirmed the presence of an OXA-48-like carbapenemase. 

*E. coli* O103A9 displayed resistance to expanded-spectrum cephalosporins (ESCs) and exhibited a synergy image in the DD-test, indicating the presence of extended-spectrum beta-lactamases (ESBLs).

The *Aeromonas* spp. were susceptible to all the antibiotics tested except for two that were resistant to cotrimoxazole.

Two coagulase-negative isolates (*S. epidermidis* and *S. pasteuri*) were considered methicillin-resistant as they were resistant to oxacillin and cefoxitin. High resistance rates to tetracycline were observed (Figure 2). The two isolates of *Bacillus* spp. were pan-susceptible to all the antibiotics tested. 

### 3.4. Resistome, MLST, Plasmidome, and O-Serogroups

Twenty strains were further investigated by WGS. Gram-positive isolates were chosen based on their resistance profile such as *mecA*-positive *Staphylococci* or their likelihood to carry virulence factors such as *S. aureus* isolates. Enterobacterales were chosen based on their resistance profile, such as expanded spectrum cephalosporin- or carbapenem-resistance on randomly chosen multisusceptible *Escherichia coli* isolates to obtain an overview of their STs or virulence factors likely involved in abscess formation.

The two *K. pneumoniae* isolates O103B1 and O103B2 harbored, in addition to *bla*_OXA-48_, several β-lactam resistance genes, including the ESBL *bla*_CTX-M-15_, *bla*_OXA-1_, *bla*_SHV-187_, and *bla*_TEM-1B_ genes. Additionally, these isolates carried several other resistance genes to different antibiotic classes (Table 5).

The *E. coli* isolate O103A10 carries the carbapenemase *bla*_OXA-48_ gene, as well as *tet(B)* and *mph(B)* genes that are responsible for tetracycline and macrolide resistance, respectively (Table 5).

Furthermore, the remaining *E. coli* strain O103A9 harbored the ESBL *bla*_CTX-M-15_ and *bla*_TEM-1B_ genes, along with aminoglycoside resistance genes *aph(6)-Id* and *aph(3****″****)-Ib*, *qnrS1* for quinolones, and *sul2* for sulphonamides (Table 5). In silico MLST typing assigned the two *E. coli* O103A9 isolate to ST1706 and ST10 for O103A10, while the two *K. pneumoniae* isolates belonged to the same ST985. Several replicon-types were identified in the four MDR Enterobacterales (Table 5). The CTX-M-15-producing *E. coli* O103A9 harbored an IncY plasmid, while *E. coli* O103A10 carried several plasmids (Table 5), including Col156, IncFIA, IncFIB, and IncFII, but no IncL plasmid known to carry *bla*_OXA-48_ gene. A careful analysis of the contig carrying the *bla*_OXA-48_ gene suggested a chromosomal location, as the *E. coli* chromosomal genes are present on both sides of *bla*_OXA-48_ gene (Figure 3). In addition, electroporation experiments with plasmids extracted from *E. coli* O103A10 failed to transfer any plasmid carrying *bla*OXA-48 gene to *E. coli* Top10.

The two carbapenemase-producing *K. pneumoniae* isolates O103B1 and O103B2 also carried several plasmids, including an IncL plasmid, likely carrying *bla*_OXA-48_ gene (Table 5). Mapping the reads against the prototypical OXA-48 plasmid revealed 100% sequence coverage, suggesting the presence of the entire plasmid in these isolates [17,18].

The resistome of seven randomly chosen *E. coli* isolates revealed few resistance genes. Five isolates produced *tetA* and *tetB* genes, together with aminoglycoside resistance genes (*aadA1*, *aadA2b*, *aph(3*′)-*Ia*, *aph*(*3*″)-*Ib*, *aph(6)-Id*) in two of the seven *E. coli* isolates. Furthermore, two *E. coli* isolates expressed a sulfonamide resistance gene (sul2 and/or sul3). These *E. coli* isolates belonged to seven different STs: ST88, ST101, ST224, ST155, ST223, ST206. Only three plasmid replicon types were detected in the investigated *E. coli* isolates; these are IncFII, IncFIA, and IncFIB. Additionally, O-serogroups that could potentially pose public health concerns were also identified, including O8, O23, O37, O42, O116, O123, O144, and O153 (Table 5).

The *S. epidermidis* strain harbored several antibiotic resistance genes, including *mecA* (which encodes PLP2A conferring resistance to methicillin), *blaZ* (which encodes a narrow spectrum penicillinase), *ant(6)* and *aph(3′)* (which encode resistance to aminoglycosides), and *fosB*, *fusB*, and *tet(K)* (which encode resistance to fosfomycin, fusidic acid, and tetracycline, respectively). The *S. pasteuri* strain harbored only *mecA* and *blaZ* genes. Four isolates of *S. aureus* (O104F5, O104F7, O104F9, and O104F10) carried the *blaZ* gene, while the *tet(K)* and *erm(T)* genes were detected only in isolates O104F4 and O104F10, respectively (Table 6).

*S. epidermidis* strain belonged to ST61, and the six *S. aureus* isolates belonged to five different STs: ST700 for two isolates, ST522 for two isolates and ST398 from sheep, and ST97 and another novel allele from cattle.

### 3.5. Virulome

As these bacterial isolates were responsible for purulent infections, the search for virulence genes was of the utmost importance (Table 6 and Table 7). The ESBL-producing *E. coli* strain O103A9 was found to harbor only three known virulence genes: *ompT*, *terC*, and *yehD*, which encode an outer membrane protease, a tellurite resistance protein C, and a fimbrial protein, respectively. The *E. coli* strain-producing OXA-48 carried three genes as well, including *terC*, *csgA*, and *traT*, which encode a tellurite resistance protein C, a curli fimbriae subunit A, and a serum-resistance-associated protein, respectively. 

Among the two *K. pneumoniae* isolates, O103B2 harbored 82 virulence genes, while O103B1 had 79 genes. Both strains shared common virulence genes such as type 1 fimbriae (*fimABCDEFGHIK*) and type 3 fimbriae operon (*mrkABCDFHIJ*), enterobactin gene clusters (*entABCDEFS* and *fepABCDG*), Type IV pili (*pilW*), the T6SS-II (*impAFGHJ*) operon, the *stbABCD* operon, and the siderophore *iroAN* cluster. Additionally, both strains were found to contain the integrative and conjugative element (*ICEKp*) containing the *yersiniabactin* gene cluster (*ybtAEPQSTUX*), *irp1*, *irp2*, and *fyuA* genes. In addition to these genes, other genes were found (Table 7).

The virulome of the seven susceptible *E. coli* isolates was also investigated, and the most common factors found were *terC*, *fimH*, *nlpI*, *lpfA*, *fyuA*, *hlyF*, *iutA*, and *cvaC* (Table 7).

The WGS analysis of *S. aureus* isolates revealed that all but one (O104F6) carried the hemolysin-encoding genes *hlgA*, *hlgB*, and *hlgC*. Leucocidin genes *lukD* and *lukE* were identified in five strains. The serine protease-encoding genes *splA* and *splB* were detected in all six strains, while *splE* was detected in only four strains. Enterotoxin genes *sec* and *sel* were detected in strains O104F7 and O104F9, and the immune evasion gene *sak* was detected in three isolates (O104F4, O104F6, and O104F8), while the *scn* gene was detected in two isolates (O104F8 and O104F10) (Table 6). 

## 4. Discussion

This study revealed a concerning rate of multidrug resistance in bacteria isolated from abscesses in one slaughterhouse in Algeria. In low- to middle-income countries, the intensification of farming is on the rise, primarily driven by a scarcity of available land and the continuous growth of human populations. This intensified farming demands greater use of antibiotics to combat infectious diseases, subsequently fostering the transfer of antibiotic resistance genes (ARGs) between microbes, resulting in multidrug-resistant bacteria in livestock but also in domestic animals and quick spread of resistant micro-organisms, which have serious consequences on animal health, productivity, and food production that pose both economic and human health problems [19,20]. Livestock animals are at risk of infectious diseases, especially with pyogenic organisms that cause abscesses at various sites of the body, and consequently have an impact on productivity, fertility, and in general to livestock health [2,21,22]. Infected carcasses at slaughterhouses can be partially or completely discarded [23].

In this study, we reported 123 abscesses, including 100 from sheep and 23 from cattle. Abscesses occur commonly in sheep as ovines are more susceptible to developing them than other ruminants. It cannot be determined solely based on clinical examination and generally relies on postmortem inspection of carcasses in slaughterhouses [24,25]. The analysis of bacterial isolates from pus showed the predominance of Enterobacterales isolates, particularly *E. coli*, followed by *Staphylococcus* spp. Enterobacterales, and *E. coli* especially have been found to be major players in the formation of abscesses in animals [5,26,27], although several studies have isolated *S. aureus* from abscesses as the quintessential suppurative pathogen in large proportions [13,14,28,29]. These bacterial species isolated are zoonotic pathogens, which places public health at risk. 

The presence of ESBLs, and in particular CTX-M-15, in animals has previously been linked to the human sector before it was also detected in animals and the environment [30]. Additionally, ceftiofur, a third-generation cephalosporin, is the main cephalosporin used in veterinary medicine due to its effectiveness in treating bacterial infections in food-producing animals; for this reason, these antibiotics could provide selection pressure that favors co-selection of plasmids carrying mobile genes (transposon, integron, cassette gene) that result in carbapenem-resistant (CR) strains [31,32]. Here, we identified a CTX-M-15 producing *E. coli* ST1706, an ST that has previously been described in Japan from different pig farms [33]. 

The presence of OXA-48-carbapenemase producers is very worrying, as carbapenems remain the last-resort therapy for treating human infections caused by MDR Gram-negative and Gram-positive bacteria [31]. However, the clinical use of these antibiotics is presently at risk due to the global proliferation of β-lactamases (BLs) with the ability to degrade them, and the increase in the worldwide emergence of carbapenem-resistant organisms (CROs), which constitute a critical growing public health threat [34]. In livestock or veterinary fields, carbapenems are not licensed and have no legal indication, so their use is prohibited [32,35,36]. Nevertheless, many studies conducted in Algeria have reported CROs in livestock, companion animals, and birds [37,38,39]. 

In our study, we isolated, for the first time, OXA-48 carbapenemase-producing Enterobacterales from abscesses of farm animals. OXA-48 is the most common carbapenemase in Enterobacterales and one of the most frequently isolated around the Mediterranean rim [40]. It is most frequently detected in *K. pneumoniae* and *E. coli* but can also occur in other Enterobacterales species [40]. Both *K. pneumoniae* isolates belong to ST985, an ST type that has been isolated from various sources and geographic locations. MDR *K. pneumoniae* ST985 isolates carrying up to 16 different resistance genes, including *bla*_CTX-M-55_ gene, were isolated from rectal swab samples of dairy cows from Quetta in Pakistan [41]. In Austria, *bla*_CTX-M-15_-producing *K. pneumoniae* ST985 were isolated from river water samples that were identical to clinical isolates from Austrian hospitals [42]. Finally, *bla*_CTX-M-15_-producing *K. pneumoniae* ST985 have been involved in an outbreak in Israel in a neonatal intensive care unit [43]. The simultaneous presence of *bla*_OXA-48_ and *bla*_CTX-M-15_ genes in *K. pneumoniae* strains ST985 is of particular concern, as this ST seem to be responsible for human and animal infections, and as the combination of these two β-lactamases lead to the resistance to almost all β-lactams. Extended-spectrum β-lactamase (ESBL) and carbapenemase genes are often associated with an MDR phenotype [44], as illustrated in our study by the presence of different resistance to different classes of antibiotics.

The *Bla*_OXA-48_ gene is usually located on transferable Inc L plasmids [45]; however, there have also been reports of chromosomally located *bla*_OXA-48_ genes in some STs, such as ST38 CPE [46,47,48]. Here, we report a chromosomal localization of the *bla*_OXA-48_ gene in *E. coli* ST 10. Chromosomal insertion of resistance genes is believed to favor the stability of resistance genes in the absence of selective pressure [49,50]. 

Methicillin-resistant *staphylococci* are among the emerging pathogens that now constitute a threat to human and animal health. Due to its rapid development of antibiotic resistance in clinical settings, methicillin-resistant *S. aureus* (MRSA) is regarded as one of major life-threatening pathogens. The recent isolation of MRSA strains in several animals is thought to be one of the main factors in the spread of infection and disease in both humans and animals [51,52]. In our study, we identified a methicillin-susceptible *S. aureus* ST398, an ST spreading worldwide in animals and humans and often referred to as livestock-associated MRSA (LS-MRSA) [53,54]. 

This resistance, as our study shows, concerned coagulase-negative Staphylococci (CoNS: *S. epidermidis* and *S. pasteuri*) isolated from abscesses and harbored the *mecA* gene, indicating methicillin resistance. Methicillin-resistant coagulase-negative *staphylococci* have been less studied, but their importance as pathogens is increasing. For a long time, *S. aureus* was thought to be the predominant pathogenic *Staphylococci* species. However, recent investigations have shown the increasing role of CoNS in causing antibiotic-resistant infections [55,56]. In a previous study conducted in Algeria, which focused on unpasteurized cow’s milk, three isolated CoNS were resistant to methicillin, and all were *mecA*-positive [57]. A few studies carried out on CoNS in livestock have revealed that food-producing animals constitute a large reservoir of multiresistant CoNS [58].

Virulence factors of Enterobacterales or *Staphylocci* are based on their ability to adhere to host cells, produce toxins, and resist host immune defenses [59]. The identification of specific virulence genes is crucial for understanding the pathogenic potential and for the development of preventive strategies, effective vaccines, and novel therapeutics [60]. For Enterobacterales, important virulence factors, including adhesins, fimbriae, intimin, capsules, iron metabolism, siderophores, heme/hemoglobin transport proteins, and cell invasion, were identified. Furthermore, a study conducted in Western Algeria in 2017 to determine the prevalence of carbapenemase-producing Enterobacteriaceae (CPE) in chicken meat revealed that carbapenemase-producing *K. pneumoniae* isolates harbored several virulence factors, such as fimH type 1 fimbriae virulence gene, ureA, involved in the hydrolysis of urea to ammonia, mrkD, encoding a type 3 fimbriae that promotes biofilm development, uge, that codes for a UDP galacturonate 4-epimerase, and wabG, encoding the biosynthesis of the core lipopolysaccharide [61]. The relationship between antimicrobial resistance and virulence factors in bacteria is complex. There are common characteristics shared between virulence and resistance, such as the involvement of efflux pumps, porins, and cell wall alterations. Additionally, some studies have found a significant association between certain virulence genes and antimicrobial resistance, suggesting that acquiring resistance to some antibiotics may impact the expression of virulence factors [62,63,64]. In contrast to other studies, KPC-producing Enterobacterales, such as *K. pneumoniae*, typically exhibit lower levels of virulence compared to non-carbapenemase-producing strains. Overall, the presence of carbapenemase genes does not necessarily imply a reduction in virulence, although there might be trade-offs between virulence and resistance capabilities in certain strains [65].

## 5. Conclusions

The presence of *bla*_ESBL_, *bla*_OXA-48_, and *mec*A genes in animal abscesses highlights the importance of monitoring the use of antimicrobial in animals. Preventing the spread of multidrug-resistant bacteria in livestock animals should, therefore, be a priority for public health, which can be achieved through the reduction in and proper use of antimicrobial agents in animal husbandry and in humans, and also acting at the farm level, improving hygiene and biosecurity measures, based primarily on the elimination of risk factors and vaccination of small ruminants.

## Figures and Tables

**Figure 1 microorganisms-12-00524-f001:**
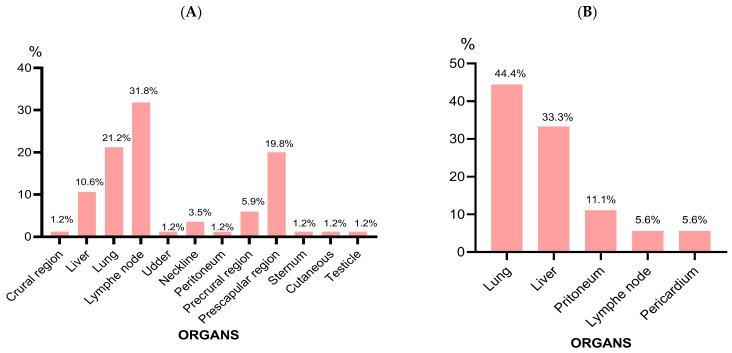
Abscess seats location in sheep (**A**) and cattle (**B**).

**Figure 2 microorganisms-12-00524-f002:**
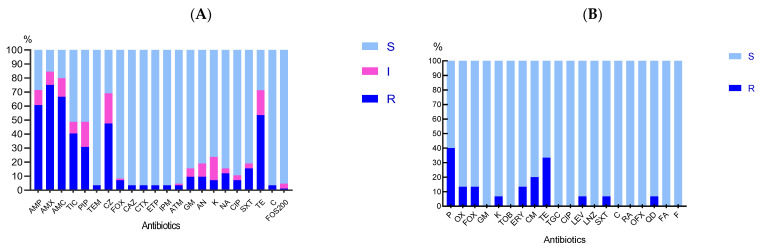
Susceptibility results of bacteria isolated from the abscesses: (**A**) Enterobacterales; (**B**) *Staphylococci.* Results were interpreted according to Eucast breakpoints 2022. (S: sensitive, I: intermediate, R: resistance). Amikacin (AN), amoxicillin (AMX), amoxicillin-clavulanic acid (AMC), ampicillin (AMP), aztreonam (ATM), cefazolin (CZ), cefotaxime (CTX), cefoxitin (FOX), ceftazidime (CAZ), chloramphenicol (C), ciprofloxacin (CIP), clindamycin (CM), ertapenem (ETP), erythromycin (ERY), fosfomycin (FOS200), fusidic Acid (FA), gentamicin (GM), imipenem (IPM), kanamycin (K), Levofloxacin (LEV), Linezolid (LNZ), nalidixic acid (NA), Nitrofurantoin (F), ofloxacin (OFX), Oxacillin (OX), penicillin (P), piperacillin (PIP), Quinupristin/dalfopristin (QD), Rifampin (RA), Temocillin (TEM), tetracycline (TE), ticarcillin (TIC), Tigecycline (TGC), Tobramycin (TOB), and trimethoprim/sulfamethoxazole (SXT). Disk loads are indicated in Section 2.

**Figure 3 microorganisms-12-00524-f003:**
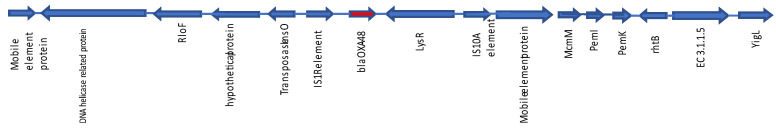
Genetic environment of *bla***_OXA-48_** gene located on the chromosome in *E. coli* O103A10. *Bla*_OXA-48_ gene is indicated in red.

**Table 1 microorganisms-12-00524-t001:** Frequency of abscesses in relation to age and sex in sheep and cattle.

	Sheep		Cattle		
Age	≤12 Months	>12 Months	≤ 2 Years	2−5 Years	>5 Years
Male	97	/	18	/	1
Female	/	3	/	3	1

**Table 2 microorganisms-12-00524-t002:** Clinical presentations of abscesses according to the bacterial species.

Pus Characteristics		*Staphylococcus aureus*	*Staphylococcus* *Coagulase negative*	*Aeromonas* spp.	*Bacillus* spp.	*Enterobacterales*
	Bacteria					
Consistency		Grumbling viscous Homogeneous viscous	Homogeneous viscous Homogeneous fluid	Homogeneous viscous Thick homogeneous	Homogeneous viscous Thick homogeneous	Grumbling fluid Grumbling viscous Homogeneous fluid Homogeneous viscous Thick grumbling Thick homogeneous Viscous grumbling hemorrhagic
Color		White Light yellow	Light yellow Yellow white Green	Green White Yellow	Green Yellow	Green White Yellow
Odor		Fade	Fade	Fade	Fade	Nauseating Fade

**Table 3 microorganisms-12-00524-t003:** Bacterial species isolated from abscesses in cattle and sheep.

Bacterial Species	Animal Species		Total	Percentage (%)
	Sheep	Cattle		
*Escherichia coli*	56	14	70	61.3%
*Staphylococcus aureus*	5	2	7	6.1%
*Aeromonas veronii*	3	3	6	5.2%
*Aeromonas hydrophila*	1	3	4	3.5%
*Klebsiella pneumoniae*	3		3	2.6%
*Morganella morganii*	2	1	3	2.6%
*Staphylococcus epidermidis*	3		3	2.6%
*Klebsiella oxytoca*	2		2	1.7%
*Citrobacter brackii*	1		1	0.9%
*Citrobacter freundii*	1		1	0.9%
*Enterobacter* spp.	1		1	0.9%
*Serratia marcescens*	1		1	0.9%
*Proteus vulgaris*	1		1	0.9%
*Lelliottia* spp.	1		1	0.9%
*Staphylococcus lentus*	1		1	0.9%
*Staphylococcus cohnii*	1		1	0.9%
*Staphylococcus simulans*	1		1	0.9%
*Staphylococcus pasteuri*	1		1	0.9%
*Staphylococcus vitulinus*	1		1	0.9%
*Aeromonas bestiarum*	1		1	0.9%
*Aeromonas salmonicida*	1		1	0.9%
*Aeromonas eucrenophila*	1		1	0.9%
*Bacillus cereus*	1		1	0.9%
*Bacillus mojavensis*	1		1	0.9%
**Total**	**91**	**23**	**114**	**100%**

**Table 4 microorganisms-12-00524-t004:** MICs of Enterobacterales harboring *bla*_OXA-48_ gene.

Antimicrobial (s)	*K. pneumoniae* O103B2 *	*K. pneumoniae* O103B1 *	*E. coli* O103A10 *
Amoxicillin	>32 (R)	>32 (R)	>32 (R)
Amoxicillin + CLA	>128 (R)	>128 (R)	>128 (R)
Ticarcillin	>32 (R)	>32 (R)	<4 (S)
Piperacillin	>32 (R)	>32 (R)	<4 (S)
Piperacillin-tazobactam	>32 (R)	>32 (R)	<4 (S)
Temocillin	256(R)	256 (R)	512 (R)
Tigecycline	1 (R)	1 (R)	1 (R)
Ceftazidime	16 (R)	16 (R)	0.25 (S)
Ceftazidime/Avibactam	0.25 (S)	0.25 (S)	0.25 (S)
Cefotaxime	8 (R)	8 (R)	0.5 (S)
Ceftolozane/Tazobactam	16 (R)	16 (R)	0.5 (S)
Cefepime	8 (I)	8 (I)	0.5 (S)
Cefiderocol	1 (S)	1 (S)	0.12 (S)
Aztreonam	16 (R)	16 (R)	0.12 (S)
Imipenem	1 (S)	1 (S)	0.5 (S)
Imipenem/Relebactam	0.5 (S)	0.5 (S)	0.25 (S)
Meropenem	0.5 (S)	0.5 (S)	0.12 (S)
Meropenem/Vaborbactam	0.5 (S)	0.5 (S)	0.12 (S)
Ertapenem	2 (R)	2 (R)	0.5 (S)
Ciprofloxacin	2 (R)	2 (R)	0.12 (S)
Tobramycin	8 (R)	8 (R)	1 (S)
Levofloxacin	0.5 (S)	1 (I)	0.25 (S)
Colistin	0.5 (S)	0.5 (S)	0.5 (S)

* Values are in micrograms per milliliter. S and R stand for susceptible and resistant.

**Table 5 microorganisms-12-00524-t005:** Resistance genes, O-serogroups, MLST, plasmid replicons, and virulence factors detected in *E. coli* and *K. pneumoniae* isolates as revealed with ResFinder-4.1, SerotypeFinder-2.0, MLST 2.0, PlasmidFinder 2.0, and VirulenceFinder-2.0 softwares available at CGE the center for genomic epidemiology. (https://cge.food.dtu.dk/ (accessed on 24 January 2024)) and virulence factor database (VFDB) (http://www.mgc.ac.cn/VFs/main.htm (accessed on 20 December 2023)).

Isolates ^1^	Clinical Features			MLST	Serotype	Antibiotic Resistance Genes																								Point Mutations	Plasmid Replicon									
						Beta-Lactam								Aminoglycosides				Quinolones		Various																				
	Animal ^2^	Site ^3^	Date of Isolation ^4^			*bla* _CTX-M-15_	*bla* _SHV-187_	*bla* _TEM-1B_	*bla* _TEM-1C_	*bla* _OXA-48_	*bla* _OXA-1_	*aadA1*	*aadA2b*	*aac(6′)-Ib-cr*	*aph(3′)-Ia*	*aph(6)-Id*	*aph(3″)-Ib*	*qnrB1*	*qnrS1*	*sul2*	*sul3*	*dfrA14*	*fosA*	*tet(A)*	*tet(B)*	*catB3*	*mph(B)*	*sitABCD*	*cmlA1*		IncY	Col	Col156	IncFIA	IncFIB	IncFII	ColRNAI	IncFIB(K)	IncFII(K)	IncL
*Ec O103A9*	S	PR	*23/04*	ST1706	O116:H8																									acrR parC										
*Ec O103A10*	S	LN	*12/03*	ST10	O8:H10																																			
*Kp O103B1*	S	PR	*28/05*	ST985																																				
*Kp O103B2*	S	LN	*29/05*	ST985																										acrR ompK36 ompK37										
*Ec O104G3*	C	Li	*21/05*	ST88	O8:H17																																			
*Ec O104G4*	S	Li	*16/06*	ST101	O153:H21																																			
*Ec O104G5*	S	Li	*21/05*	ST224	O42:H8																									gyrA parE parC										
*Ec O104G6*	C	PR	*09/03*	ST155	O23:H21																																			
*Ec O104G7*	S	L	*22/05*	ST223	O:123:H21																																			
*Ec O104G8*	S	PR	*06/03*	ST206	O144:H5																									parC										
*Ec O104H4*	C	L	*28/05*	Unknown	O37:H12																																			

(1) Ec: *E. coli*; Kp: *K. pneumoniae*; (2) S: sheep; C: cattle; (3) PR: prescapular region; LN: lymph node; Li: liver; L: lung; (4) day/month/2019. Filled boxes indicated presence of a given allele.

**Table 6 microorganisms-12-00524-t006:** Resistance genes, MLST, and virulence factors detected in *Staphylococci* isolates, as revealed with ResFinder-4.1, MLST 2.0, PlasmidFinder 2.0, and VirulenceFinder-2.0 softwares available at CGE the center for genomic epidemiology. (https://cge.food.dtu.dk/ (accessed on 24 January 2024)).

Isolates ^1^	Clinical Features			MLST	Antibiotic Resistance Genes								Virulence Factors																					
					Beta-lactams		Aminoglycosides		Fosfomycin	Fusidic acid	Tetracycline	Erythromycin	Adherence				Enzyme											Immune Evasion			Toxin			
	Animal ^2^	Site ^3^	Date of Isolation ^4^		*bla* _Z_	*mecA*	*ant(6)*	*aph(3′)*	*fosB*	*fusB*	*tet(K)*	*erm(T)*	*ebp*	*sdrC*	*sdrG*	*icaA,B,C*	*hlg*	*luk D,E*	*sak*	*spl*	*sspA*	*sspB*	*sspC*	*geh*	*lip*	*aur*	*nuc*	*capB*	*scn*	*hlb*	*sec*	*sea*	*sel*	*cylR2*
*Sepi O104F3*	S	PR	*23/04*	ST61																														
*Spas O104G1*	S	NK	*21/05*																															
*Sa O104F4*	S	LN	*18/06*	ST522																														
*Sa O104F5*	C	Li	*21/04*	Unk																														
*Sa O104F6*	S	NK	*24/02*	ST522																														
*Sa O104F7*	S	LN	*06/03*	ST700																														
*Sa O104F8*	C	L	*29/05*	ST97																														
*Sa O104F9*	S	LN	*05/03*	ST700																														
*Sa O104F10*	S	LN	*05/05*	ST398																														

(1) Sa: *S. aureus*; Sepi: *S. epidermidis*; Spas: *S. pasteuri* (2) S: sheep; C: cattle; (3) PR: prescapular region; LN: lymph node; Li: liver; NK: neckline; L: lung; (4) day/month/2019. Filled boxes indicated presence of a given allele.

**Table 7 microorganisms-12-00524-t007:** Virulome of *E. coli* and *K. pneumoniae* isolates.

				Virulence Factors																																																	
Bacteria ^1^	Animal ^2^	Site ^3^	MLST	Various																									Hemolysin	Type 3 Fimbriae	Type 1 Fimbriae	Type IV Pili	Ent Siderophore					Salmochelin		Aerobactin		Yersiniabactin			Two-component Regulatory System		T6SS	Stb	Ferrous iron Transport			AcrAB	
				*fimH*	*F17 A,C,D,G*	*ompT*	*terC*	*csgA*	*traT*	*traJ*	*cia*	*cvaC*	*etsC*	*gad*	*lpfA*	*mchF*	*mchC*	*nlpI*	*papC*	*tia*	*yeh A,B,C,D*	*cma*	*espI*	*iha*	*iss*	*hra*	*AslA*	*colE2- like*	*hlyF*	*mrkA-J*	*fimA-K*	*pilW*	entA-F	entS	fepA-D	fepE	fes	*iroE*	*iroN*	*iutA*	*iucC*	irp1-2	ybtA, E, Q, S, T, U, X	fyuA	*rcsA*	*rcsB*	23 genes	*stbA-D*	*sit A*	*sitD*	*sitC*	*AcrAB*	*acrB*
*Ec O103A9*	S	PR	*ST1706*																																																		
*Ec O103A10*	S	LN	*ST10*																																																		
*Kp O103B1*	S	PR	*ST985*																																																		
*Kp O103B2*	S	LN	*ST985*																																																		
*Ec O104G3*	C	Li	ST88																																																		
*Ec O104G4*	S	Li	ST101																																																		
*Ec O104G5*	S	Li	ST224																																																		
*Ec O104G6*	C	PRC	ST155																																																		
*Ec O104G7*	S	L	ST223																																																		
*Ec O104G8*	S	PR	ST206																																																		
*Ec O104H4*	C	L	Unk																																																		

(1) Ec: *E. coli*; Kp: *K. pneumoniae*; (2) S: sheep; C: cattle; (3) PR: prescapular region; LN: lymph node; Li: liver; PRC: pericardium; L: lung. Filled boxes indicated presence of a given allele.

## Data Availability

Raw data are available upon request, and WGS sequences have been deposited at NCBI.

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
