# Peer review of "Antibiotic Susceptibility Profiles of Bacterial Isolates Recovered from Abscesses in Cattle and Sheep at a Slaughterhouse in Algeria"

_microorganisms, 2024, doi:10.3390/microorganisms12030524_

Round 1
Reviewer 1 Report
Comments and Suggestions for Authors
This manuscript aims to explore the epidemiology of abscesses in sheep and cattle, focusing on the distribution, characteristics, and bacterial species responsible for these abscesses. The study involves the examination of 123 abscess samples from slaughtered sheep and cattle, identifying 114 bacterial strains from 103 abscesses and performing antimicrobial susceptibility testing.
Major Concerns:
-
Objective and Results Discrepancy: The manuscript's stated objective is to identify bacteria responsible for abscess formation in sheep and cattle. However, the study merely lists all separated strains without clearly establishing their role in abscess formation. The relevance of each identified bacterial species to abscess formation needs to be more explicitly addressed.
-
Methodological Limitations: The exclusive use of MALDI-TOF for bacterial identification is a significant limitation. To ensure accuracy, it is advisable to complement this with more precise methods such as 16S rRNA gene sequencing or whole-genome sequencing.
-
Title and Content Inconsistency: The emphasis on Enterobacteriaceae in the manuscript's title is misleading, as this is not reflected in the introduction or adequately justified in the content. A title that more accurately reflects the study's focus is recommended.
-
Presentation of Results: The presentation of antimicrobial susceptibility results in Figure 2 requires improvement for clarity. Specifically, labeling the meaning of "S," "I," and "R," and employing distinct colors to differentiate between categories of antimicrobial drugs, would significantly enhance reader comprehension.
Comments on the Quality of English Language
Minor editing of English language required
Author Response
Please find below our answers to your queries
-Objective and Results Discrepancy: The manuscript's stated objective is to identify bacteria responsible for abscess formation in sheep and cattle. However, the study merely lists all separated strains without clearly establishing their role in abscess formation. The relevance of each identified bacterial species to abscess formation needs to be more explicitly addressed.
Authors’ answer: In most of the cases, only the identified bacteria grew from the abcesses. The characteristics of these abscesses have now been listed in Table 2.
Methodological Limitations: The exclusive use of MALDI-TOF for bacterial identification is a significant limitation. To ensure accuracy, it is advisable to complement this with more precise methods such as 16S rRNA gene sequencing or whole-genome sequencing
Author’s answer: The authors agree with the reviewer’s comments, however sequencing of all isolates would have required significant resources in terms of time, costs and data analysis, far surpassing the limits of our study. We have instead used MALDI-TOF, which is the method used in most clinical and veterinary bacteriology laboratories as it is recognized as a reliable and efficient method for bacterial identification. It offers a whole range of advantages, including speed, high precision, and efficiency. All the bacterial identification yielded scores above 2.2.
Presentation of Results: The presentation of antimicrobial susceptibility results in Figure 2 requires improvement for clarity. Specifically, labeling the meaning of "S," "I," and "R," and employing distinct colors to differentiate between categories of antimicrobial drugs, would significantly enhance reader comprehension.
Authors’ answer: Corrected accordingly. (Line 243).
Reviewer 2 Report
Comments and Suggestions for Authors
Authors evaluated the AMR, virulence, and genetic profiles of 114 bacterial strains isolated from abscesses of cattle and sheep. Interesting and worrying data about AMR profiles of circulating strains emerged in this study, hihgliting the need for a better control and monitoring of abscesses both at farm and slaughterhouse level.
Although the study is of scientific interest and add valuable information about antimicrobial resistance of relevant bacteria in cattle and sheep, it seems a little confusing and lacks a structured and organic flow.
Below, I included some comments and suggestions.
Overall evaluation:
The title, results, and conclusion, seem to belong to three different studies. Authors should try to organize data to give to the manuscript an organic flow.
Tables should be better in a better format (e.g. deleting vertical and horizontal lines, maintaining only top and bottom lines). Please, pay attention to caption format, they are all different (bold/non bold).
Moreover, Authors should evaluate to move some tables in “Supplementary Materials” (e.g. Tables 4 and 5).
Title: It does not reflect the study. Author should consider to use a fitter title (e.g. “Antimicrobial resistance of bacteria isolated from….”)
Introduction:
Line 43: Authors should briefly introduce the common causes of abscesses in cattle and sheep.
Line 72: what does it mean “zoonotic profile of microorganism”?. Please rephrase the sentence.
M&M:
Lines 79-86: Usually the number of tested animals should be stated in this section. I suggest Authors to move from Results to M&M. Authors should also state here the localization of examined abscesses.
Line 107: Please, add “order” in this sentence for Enterobacterales (e.g. bacteria belonging to Enterobacterales order).
Results:
Authors should state which bacteria species belonging to Enterobacterales have been isolated.
Line 199: Please, revise “Bacillus” in italic.
Line 253: 3.4 section is confusing and difficult to follow. Please, revise.
Table 1:
There is some french text. Please revise.
Conclusions:
Some sentences of this section are off-topic, please consider to revise. Authors should consider to include here the importance of control strategies for reducing the presence of abscesses in farm animals and to contrast AMR.
Author Response
Please find below our answers to your queries
-Overall evaluation:
The title, results, and conclusion, seem to belong to three different studies. Authors should try to organize data to give to the manuscript an organic flow.
Tables should be better in a better format (e.g. deleting vertical and horizontal lines, maintaining only top and bottom lines). Please, pay attention to caption format, they are all different (bold/non bold).
Moreover, Authors should evaluate to move some tables in “Supplementary Materials” (e.g. Tables 4 and 5).
Authors’ answer: Corrected accordingly. We feel that tables 4 and 5 are important and we prefer to keep them in the main article.
-Title: It does not reflect the study. Author should consider to use a fitter title (e.g. “Antimicrobial resistance of bacteria isolated from….”)
Authors’ answer: The title was modified accordingly.
- Introduction:
Line 43: Authors should briefly introduce the common causes of abscesses in cattle and sheep.
Authors’ answer: Corrected accordingly (Lines 44-53).
Line 72: what does it mean “zoonotic profile of microorganism”?. Please rephrase the sentence.
Authors’ answer: The term "zoonotic profile of microorganisms" refers to the characteristics or properties of microorganisms that enable them to infect and be transmitted between animals and humans, causing what are known as zoonotic diseases. Zoonotic diseases are those that can be transmitted naturally between animals and humans.
The sentence containing “zoonotic profile of microorganism” was modified accordingly (see Line 80).
-M&M:
Lines 79-86: Usually, the number of tested animals should be stated in this section. I suggest Authors to move from Results to M&M. Authors should also state here the localization of examined abscesses.
Authors’ answer: Donne accordingly (Line 90-99).
Line 107: Please, add “order” in this sentence for Enterobacterales (e.g. bacteria belonging to Enterobacterales order).
Authors’ answer: Corrected accordingly (Line 125).
-Results:
Authors should state which bacteria species belonging to Enterobacterales have been isolated.
Line 199: Please, revise “Bacillus” in italic.
Line 253: 3.4 section is confusing and difficult to follow. Please, revise.
Authors’ answer: Corrected accordingly. (Lines 221 and 280) respectively.
Table 1: There is some french text. Please revise.
Authors’ answer: Sorry about that. It has been corrected accordingly.
-Conclusions:
Some sentences of this section are off-topic, please consider to revise. Authors should consider to include here the importance of control strategies for reducing the presence of abscesses in farm animals and to contrast AMR.
Authors’ answer: The Conclusion was Corrected and modified accordingly. (Lines 509-end).
Reviewer 3 Report
Comments and Suggestions for Authors
The article submitted by YOUSFI et al. presents the results of a microbiological evaluation of animal abscesses. The results are interesting, but refinement is needed, including in the organization of the manuscript, formatting, and English. Below are some comments that should be considered by the authors.
Introduction
I missed the authors mentioning resistance to carbapenem antibiotics, considering that it even appears in the title.
Materials and methods.
Line 82: How many abscesses were collected from how many animals? Was this procedure accompanied or carried out by a veterinarian?
Line 100: MALDI-TOF MS?
Lines 105-106: Which document did the authors use to follow the antibiotic susceptibility test? It's not very clear whether it was veterinary or clinical pathogens, which they seem to have investigated. The authors could provide the exact link that directs readers to the document that contains the antibiotics used, concentrations and breakpoints.
Lines 107-115: Why meropenem was not been tested against Enterobacterales? This antibiotic could provide interesting results.
Line 116: Aeromonas should be italic.
Line 141: It would be very interesting if the authors included the number of isolates submitted to WGS.
Line 148: De novo should be italic.
Line 148: What tools were used to assess the quality of the sequences?
Line 152: ResFinder can provide results on chromosomal mutations, for example in efflux pumps that can contribute to antibiotic resistance (ompK36, ompK37, etc). The authors need to implement this analysis.
Results
Table 1: In the table, there must be no cell without an element. I suggest putting "-" or "/".
Line 199: Bacillus should be italic.
Table 3: It is strongly recommended that authors put next to the MIC values whether the isolate was resistant, sensitive or intermediate. For example for Amoxicillin = MIC >32 (R). This would certainly help the reader to interpret the result.
Line 253: It's not at all clear what the selection criteria are for WGS. So far, I thought that the three OXA-48-producing isolates would be sequenced. When I look at Table 4, I see other isolates. What were the criteria? This needs to be very well defined so as not to generate confusion.
Line 268: E. coli does not produce the gene but carries the gene encoding the OXA-48 enzyme. Rewrite.
Lines 279-281: Was a probe hybridization experiment carried out to confirm the chromosomal location? Southern blot.
Line 286-289: What is the genetic context of the blaOXA-48 gene? Integron? Transposon? Flanked by IS?
Line: 295: sul1 gene is menttioned here, but not show in Table 4.
Table 5: Table 5 is clearly in figure format and has low resolution. The authors should provide the appropriate table.
Table 6: The same earlier commentary.
Line 336: I don't see the terC gene as a virulence factor. Currently, metal tolerance genes are widespread in different bacterial species. They basically operate in the intracellular management of metals. Please check this statement or bring references that mention terC, as well as its operon, is representative of a virulence factor.
Discussion
Lines 376-377: It would be interesting for the authors to discuss the difference in occurrence between species and sex.
Lines 408-410: This is a very delicate statement and depends a lot on the species. For example, when analyzing K. pneumoniae, blaKPC is the main carbapenemase gene detected. I think the authors need to review this information.
Lines 428-429: The authors could confirm this using XbaI-PFGE followed by probe hybridization.
General comments - Discussion: The authors did not discuss virulence factors and a One Health view would have been very well accepted.
Author Response
Please find below our answers to your queries
-Introduction I missed the authors mentioning resistance to carbapenem antibiotics, considering that it even appears in the title.
Authors’ answer: As suggested by reviewer 2, the title has been revised so that it matches to complete study.
Materials and methods.
Line 82: How many abscesses were collected from how many animals? Was this procedure accompanied or carried out by a veterinarian?
Authors’ answer: Corrected accordingly (Line 92-96).
indeed, the abscess collection procedure was carried out by a veterinarian who is the first author.
Line 100: MALDI-TOF MS?
Authors’ answer: Corrected accordingly.(Line: 119).
Lines 105-106: Which document did the authors use to follow the antibiotic susceptibility test? It's not very clear whether it was veterinary or clinical pathogens, which they seem to have investigated. The authors could provide the exact link that directs readers to the document that contains the antibiotics used, concentrations and breakpoints.
Authors’ answer: Accordingly rectified.
Lines 107-115: Why meropenem was not been tested against Enterobacterales? This antibiotic could provide interesting results.
Authors’ answer: carbapenems are not among the antibiotics prescribed in veterinary medicine, so a priori we limited our test to ertapenem and imipenem, to obtain an overview of the susceptibility of isolates to the carbapenems commonly used in human medicine. Meropenem was tested for carbapenem resistant isolates (Table 4).
Line 116: Aeromonas should be italic.
Authors’ answer: Corrected accordingly. (line 134)
Line 141: It would be very interesting if the authors included the number of isolates submitted to WGS.
Authors’ answer: Done accordingly. (line 159)
Line 148: De novo should be italic.
Authors’ answer: Corrected accordingly.
Line 148: What tools were used to assess the quality of the sequences?
Authors’ answer: De novo assembly was performed by CLC Genomics Workbench after quality trimming (Qs ≥20) with word size 34.
Line 152: ResFinder can provide results on chromosomal mutations, for example in efflux pumps that can contribute to antibiotic resistance (ompK36, ompK37, etc). The authors need to implement this analysis.
Authors’ answer: These data have now been provided, but the exact contribution of these mutations in the overall antimicrobial susceptibility is not clearly established (Table 5).
Results
Table 1: In the table, there must be no cell without an element. I suggest putting "-" or "/".
Authors’ answer: Done accordingly
Line 199: Bacillus should be italic.
Authors’ answer: Done accordingly
Table 3: It is strongly recommended that authors put next to the MIC values whether the isolate was resistant, sensitive or intermediate. For example for Amoxicillin = MIC >32 (R). This would certainly help the reader to interpret the result.
Authors’ answer: Done accordingly (Table 4).
Line 253: It's not at all clear what the selection criteria are for WGS. So far, I thought that the three OXA-48-producing isolates would be sequenced. When I look at Table 4, I see other isolates. What were the criteria? This needs to be very well defined so as not to generate confusion.
Authors’ answer: As suggested by reviewer 2, the title has been changed to represent better the entire study, and not only the three CPEs, which are of course worrying descriptions. For the WGS, we have concentrated on Gram-positive isolates that were either MDR such as mecA positive Staphylococci or S. aureus isolates known to carry virulence factors, and on enterobacterales that were either MDR or on randomly chosen multi-susceptible Escherichia coliisolates to obtain an overview of their virulence factors likely involved in in abscess formation. This information has been added to the MS (Line 280-287)
Line 268: E. coli does not produce the gene but carries the gene encoding the OXA-48 enzyme. Rewrite.
Authors’ answer: Corrected accordingly (Line 297)
Lines 279-281: Was a probe hybridization experiment carried out to confirm the chromosomal location? Southern blot.
Authors’ answer: The chromosomal location is based on indirect evidence, such as insertion of blaOXA-48 within chromosomal genes and negative conjugation and electroporation experiments. This has been clearly indicated in the text (line312-314)
Line 286-289: What is the genetic context of the blaOXA-48 gene? Integron? Transposon? Flanked by IS?
Authors’ answer: It is likely that transposition of Tn1999 is at the origin in the insertion.
Line: 295: sul1 gene is menttioned here, but not show in Table 4.
Authors’ answer: Corrected accordingly in the text.
Table 5: Table 5 is clearly in figure format and has low resolution. The authors should provide the appropriate table.
Table 6: The same earlier commentary.
Authors’ answer: Modifiable tables have been provided now
Line 336: I don't see the terC gene as a virulence factor. Currently, metal tolerance genes are widespread in different bacterial species. They basically operate in the intracellular management of metals. Please check this statement or bring references that mention terC, as well as its operon, is representative of a virulence factor.
Authors’ answer: while there is a suggestion of a relationship between tellurite resistance and pathogenicity, the specific role of TerC as a virulence factor is not definitively established. TerC is listed as a virulence factor by VirulenceFinder 2.0 (Center of genomic epidemiology) and by virulence factor database (VFDB) (http://www.mgc.ac.cn/VFs/main.htm), but we agree, further research would be needed to confirm this relationship.
Discussion
Lines 376-377: It would be interesting for the authors to discuss the difference in occurrence between species and sex.
Author’s answer: We fully agree with the reviewer’s comment. The low occurrence of abscesses in females is likely explained by the fact that females under 1 or two years of age for Sheep and cattle, respectively, are not allowed to be slaughtered. Similarly, the higher prevalence of abscesses in the ovine species as compared to the bovine species, is likely related to the lower number of cattle slaughtered per day as compared to sheep, which represent the main red meat provider in Algeria.
Lines 408-410: This is a very delicate statement and depends a lot on the species. For example, when analyzing K. pneumoniae, blaKPC is the main carbapenemase gene detected. I think the authors need to review this information.
Author’s answer: We have replaced worldwide by Mediterranean rim. For instance, in France, irrespective of the species, OXA-48 represents nearly 70% of the carbapenemases isolated in France, of which 50% come from North African countries (Tunisia, Algeria, Morocco). In K. pneumoniae, KPC represents only 2-3% of carbapenemases, even though in Italy and Greece the situation is very different, with KPC the most prevalent enzyme, with however, an rapid spread of OXA-48- and NDM-producers
Lines 428-429: The authors could confirm this using XbaI-PFGE followed by probe hybridization.
Author’s answer: The chromosomal location is suported by indirect evidence. Indeed, Pulse filed gel electrophoresis or long read sequencing (PacBio) could clearly demonstrate the chromosomal location. Unfortunately, neither of these technologies were available.
General comments - Discussion: The authors did not discuss virulence factors and a One Health view would have been very well accepted.
Authors’ answer: Corrected accordingly (Line 485,509).
Round 2
Reviewer 2 Report
Comments and Suggestions for Authors
Authors complied with all comments and suggestions. The manuscript has been improved in its quality and clarity.
Reviewer 3 Report
Comments and Suggestions for Authors
In this new version of the manuscript, the authors have made substantial changes, which have certainly improved the manuscript. All the most important points have been accepted by the authors and others have been justified. I therefore believe that the article is ready to be accepted for publication in Microorganisms. Well done!